

# Heuristic Approach to Multidimensional Temporal Assignment of Spatial Grid Points for Effective Vegetation Monitoring and Land Use in East Africa

Virginia M. Miori, Ph.D.[1], Nicolle Clements, Ph.D.[1], Brian W. Segulin[2]

[1]Department of Decision and System Sciences, Saint Joseph's University, Philadelphia, PA *19131 USA*
[2]The Rovisys Company, Aurora, Ohio 44202 USA

*Correspondence to*: Virginia M. Miori (vmiori@sju.edu)



**Abstract:** In this research, vegetation trends are studied to give valuable information toward effective land use in the East African region, based on the Normalized Difference Vegetation Index (NDVI). Previously, testing procedures controlling the rate of false discoveries were used to detect areas with significant changes based on square regions of land. This paper improves the assignment of grid points (pixels) to regions by formulating the spatial problem as a multidimensional temporal assignment problem. Lagrangian relaxation is applied to the problem allowing reformulation as a dynamic programming problem. A recursive heuristic approach with a penalty/reward function for pixel reassignment is proposed. This combined methodology not only controls an overall measure of combined directional false discoveries and nondirectional false discoveries, but make them as powerful as possible by adequately capturing spatial dependency present in the data. A larger number of regions are detected, while maintaining control of the mdFDR under certain assumptions.

Data Link: https://figshare.com/s/ed0ba3a1b24c3cb31ebf

DOI:

https://figshare.com/articles/NDVI_and_Statistical_Data_for_Generating_Homogeneous_Land_Use_Recommendations/5897581

**Keywords**: Land Use, Mathematical Programming, Dynamic Programming, Multiple Testing, Spatial Data and Analysis, False Discovery Rate

**1 Introduction**

Analysis of vegetation life cycles is fundamental in monitoring and planning agricultural endeavors and optimizing land use. In particular, gaining knowledge of current vegetation trends and using them to make accurate predictions is essential to minimize times of food scarcity and manage the consumption of natural resources in underdeveloped countries. Needing to understand the Earth's ecology and land cover is increasingly important as the impacts of climate change start to affect animal, plant, and human life. Vegetation trends are also closely related to sustainability issues, such as management of conservation areas and wildlife habitats, precipitation and drought monitoring, improving land usage for livestock, and finding optimum agriculture seeding and harvest dates for crops.

For this reason, there are many agencies and organizations that focus on the study of land use and land cover trends, linking them to climate change and the socioeconomic consequences of these changes. The United States Global Change Research Program (Land Use and Land Cover Change Interagency Working Group), the United Nations Framework Convention on Climate Change (Land Use, Land Use Change, and Forestry), and NASA's Land Cover Land Use Change Program are just three examples of well-known interdisciplinary/ interagency programs that conduct and sponsor research related to the question of global land change as noted in OCHA (2011).

Assessment of changes in a region's vegetation structure is challenging, especially in topographically diverse areas, like East Africa. Forecasting future vegetation and agricultural planning become particularly difficult when



unknown trends are occurring. However, the regions with vegetation changes are often the areas of most interest in
land use management.  Ideally, an automated screening process can identify areas with significant vegetation
changes and facilitate objective decision making about land-use management such as in Cressie & Wikle (2011).
As a first step in creating an automatic screening processes, data collection on vegetation and land cover is needed.
This is typically done through satellite remote sensing. The remote sensing imagery is used to convert the observed
elements (i.e., the image color, texture, tone, and pattern) into numeric quantities at each pixel in the image. The
image pixels correspond to a square grid of land, the size of which depends on the satellite's resolution. One such
numeric indicator is the normalized difference vegetation index (NDVI). In this article, the NDVI series came from
satellite remote sensing data collected between 1982 and 2006 over 8,000-meter grid points.  It has been shown to be
highly correlated with vegetation parameters such as green-leaf biomass and green-leaf area, and hence is of
considerable value for vegetation monitoring as in Curran (1980) and Jackson, Et al. (1983).
The NDVI standard scale ranges from −1 to 1, indicating how much live green vegetation is contained in the
targeted pixel. An NDVI value close to 1 indicates more abundant vegetation.  For example, low values of NDVI
(say, 0.1 and below) correspond to scarce vegetation consisting mostly of rock, sand and dirt. A range of moderate
values (0.2 to 0.3) indicates small vegetation such as shrub or grassland; larger NDVI values can be found in
rainforests (0.6 to 0.8). Often, negative NDVI values are consolidated to be zero since negative values indicate non-
vegetation and are of little use for vegetation monitoring.  Vegetation activity is a continuous space-time process and
NDVI data provide a space-time lattice system, in the sense that observations are available over equally spaced
regular grids. Often, the spatial resolution ranges from 1000 to 8000 meters, while the temporal one ranges from 7
days to 1 month.
Statistical and computational methods are needed to analyze remotely sensed data, like NDVI values, to determine
trends in land condition and to predict areas at risk from degradation.  Methodologies that detect land cover changes
need to be sensitive as well as accurate, since it can be costly and risky to relocate human populations, agriculture or
livestock to new regions of detected change. In such spatio-temporal data, time series models are tempting for
representing such processes. Other existing change detection methodologies include the geographically weighted
regression of Foody (2003), the principal component analysis of Hayes & Sader (2001), and the smoothing
polynomial regression of Chen & Tamura (2004). However, these methods are unable to provide an upper bound on
false detections. Since there is large risk associated with falsely declaring an area to have significant vegetation
changes, land use managers seek new methods that have a meaningful control over such errors.
In this article, we build on the previous work of Vrieling, et al. (2008) and Clements, et al., (2014).  Vrieling, et al.
(2008) first investigated this vegetation screening problem in the hypothesis testing framework of but did not
attempt to address the inherent multiplicity issue by controlling an overall false detection rate while making their
final conclusions. Clements, et. al. (2014) made improvements by incorporating the spatial dependencies, somewhat
arbitrarily, before applying multiple testing procedures.  The arbitrary spatial dependency was accounted for by
dividing the region into square blocks, based on an overall measure of spatial correlation using a semivariogram





plot. After creating such sub-regions, two-sided monotonic trend tests from Brillinger (1989) were used to identify
significant increasing or decreasing monotonic vegetation changes based on these arbitrarily chosen square regions
of land. They demonstrated that this screening procedure controlled the mixed directional false discovery rate
(mdFDR), which is defined as the expected proportion of Types I errors (False Positives) and Type III errors
(Directional errors) among all rejected null hypotheses, introduced by Benjamini & Yekutieli (2005).
In this article, we utilize the same historic NDVI time series for East Africa from 1982 to 2006. Since real-time
monitoring for change is not part of the scope, we focused improving the methodologies previously used to identify
significant changes in land cover in the region. We do this by first framing the research question as an NP-hard
temporal multi-objective assignment problem. Using heuristics to solve this problem, we first find improved sub-
regions than the previous arbitrarily chosen square grids. Using this approach allows us to adequately capture the
specific data structure and answer questions in the present context. Secondly, we reapply the multiple testing
procedures in Clements, et al., (2014) and demonstrate that the testing procedure become more powerful while still
maintaining control an error rate, the mdFDR. In summary, our methods aim to incorporate spatial local
dependencies using a multi-dimensional assignment problem formulation to improve sub-region formation, which in
turn improves the multiple testing results.
We organize the paper as follows. In the next section, we give a review of the literature followed by a detailed
description of the historical data set. We then describe the temporal assignment problem formulation to create more
homogeneous sub-regions and explain the heuristic procedure using dynamic programming. Next, we apply the
multiple testing procedures to the improved sub-regions. Finally, we reveal the results of the model implementation,
followed by a discussion, conclusions, and final remarks.
**2 Literature**
**2.1 Multiple Testing Overview**
To control over false vegetation trend detections, multiple testing procedures can be employed. An overview of
multiple testing notation and procedures are described next. When testing a single null hypothesis against a two-
sided alternative, two types of error can occur when a directional decision is made following rejection of the null
hypothesis. These are Type I error and Type III (or directional) errors. The Type I error occurs when the null
hypothesis is falsely rejected, while the Type III error occurs when the null hypothesis is correctly rejected but a
wrong directional decision is made about the alternative.
Consider testing n hypotheses simultaneously, such as testing for trend changes in n pixels over the East African
region. Table 1 gives the various outcomes of these tests, where $H_{i0}: \theta_i = \theta_{i0}$ is the null hypothesis and $H_{i1}: \theta_i \neq$
$\theta_{i0}$ is the two-sided alternative, for $i = 1, 2, \ldots, n$. Of these quantities in Table 1, only $n, A,$ and $R$ (where $R = R_1 +$
$R_2$) are known after applying a particular testing procedure. The number of Type I errors, Type II errors, and Type
III errors are $V = V_1 + V_2$, $T = T_1 + T_2$, and $U = S_2 + S_3$ respectively. All three quantities are unknown but



desirably small.  Most multiple testing procedures focus on controlling V in some capacity. In this paper, we utilize a
procedure that controls V and $U$.

| Truth | | Decision | | | Total |
|---|---|---|---|---|---|
| | | *Fail to Reject Null* | *Reject Null $H_0^{(+)}$* | *Reject Null $H_0^{(-)}$* | Total |
| | $\theta = \theta_0$ | $W$ (Correct Decisions) | $V_1$ (Type I errors) | $V_2$ (Type I errors) | $n_0$ |
| | $\theta > \theta_0$ | $T_1$ (Type II errors) | $S_1$ (Correct Decisions) | $S_2$ (Correct Decisions) | $n_+$ |
| | $\theta < \theta_0$ | $T_2$ (Type II errors) | $S_3$ (Correct Decisions) | $S_4$ (Correct Decisions) | $n_-$ |
| | Total | $A$ | $R_1$ | $R_2$ | $n$ |


119                          **Table 1: Multiple Testing outcomes from testing n hypotheses**

One of the most commonly used measures of overall Type I error is called the Familywise Error Rate (FWER).  The
FWER is the probability of making one or more Type I errors.  In other words, out of n simultaneously tested
hypotheses, where V is the number of Type I errors made out of n decisions (recall: V is an unknown quantity), then
FWER =Prob{$V > 0$}.  In the case of multiple hypothesis testing, the FWER should be controlled at a desired
overall level, called α.  The Bonferroni procedure is the most popular method to control the FWER, but there are
other techniques, such as those in Holland & Copenhaver (1987), Hochberg & Tamhane (1987), Šidák (1967), Holm
(1979), Hochberg (1988), Sarkar (1998), and Sarkar & Chang (1997).
The False Discovery Rate (FDR), proposed by Benjamini and Hochberg (1995), is the second most common
measure of Type I errors.  The FDR is the expected proportion of Type I errors among all the rejected null
hypotheses. If there are no rejected hypotheses, the FDR is defined to be zero.  In terms of Table 1, FDR =
$E\left[V/_{\max(R, 1)}\right]$. Comparatively, the FDR is less conservative than the FWER, meaning FWER control ensures
FDR control. However, a multiple testing procedure with FDR control will not necessarily maintain control of the
FWER.  The FDR is a widely accepted and utilized notion of Type I errors in large-scale multiple testing
investigations.  Recent literature has proposed methods to control the FDR, including Benjamini and Hochberg
(1995), Benjamini and Yekutieli (2001), Sarkar (2002), Blanchard and Roquain (2009), Storey, Taylor, and
Siegmund (2004), and Benjamini, Krieger, and Yekutieli (2006).
Often, it becomes essential for researchers to determine the direction of significance, rather than significance alone,
when testing multiple null hypotheses against two-sided alternatives. In other words, for each test, researchers have
to decide whether or not the null hypothesis should be rejected and, if rejected, determine the direction of the
alternative. Typically, this direction is determined based on the test statistic falling in the right- or left-side of the
rejection region. Such decisions can potentially lead to one of two types of error for each test resulting in rejection of
the null hypothesis - the Type I error if the null hypothesis is true or the directional error, also known as the Type III



error, if the null hypothesis is not true but the direction of the alternative is falsely declared (i.e. a rejection of a false
null using a two-sided alternative, but where the sign of the true parameter, say $\beta_i$, is opposite of its estimate $\widehat{\beta}_i$).
Two variants to deal with Type I and Type III errors have been introduced in the literature. First is the pure
directional FDR (dFDR), which is the expected proportion of directional errors among rejected hypotheses. Second
is the mixed directional FDR (mdFDR), which is the expected proportion of Type I and Type III errors among
rejected hypotheses. To deal with both errors in an FDR framework, the notion of mixed directional FDR (mdFDR)
was been introduced by Benjamini et al. (1993).  Since then, other methods to control directional errors have been
introduced, including Benjamini and Yekutieli (2005), Benjamini and Hochberg (2000), Shaffer (2002), Williams et
al. (1999), Guo et al. (2009), and Sarkar and Zhou (2008).
Controlling both false discoveries (V, from Table 1) and directional false discoveries (U, from Table 1) is important
in this application. For instance, when declaring a particular 8,000 m × 8,000 m grid of land as 'significantly'
changing in terms of vegetation, a Type I error is made if the area is not truly changing, and a Type III error is made
if the area is truly changing but in the opposite direction of what is determined from the data. When such decisions
are made simultaneously based on testing multiple hypotheses, one should adjust for multiplicity and control an
overall measure of Types I and III errors. Without such multiplicity adjustment, more Types I and III errors can
occur than the desired α level. It is particularly important to avoid these errors as much as possible in the present
application. Land use managers, government and local farmers are looking to relocate East African populations of
people, livestock and crops to areas of promising vegetation changes and avoid regions with decreasing changes.
Since these migrations can be risky and costly, a careful consideration of the multiplicity issue seems essential when
making declarations of significant vegetation changes.
In this article, p-values generated using the monotonic trend test in Brillinger (1989) are computed for each site
(8,000 m × 8,000 m grid of land) and provide evidence of vegetation change occurring over the years—the smaller
the p-value, the higher is the evidence of a significant vegetation change. For each site, a decision must be made
regarding the significance of vegetation change that might have occurred over the years at that site, and, if
vegetation change is found significant, determine the direction in which this change has taken place. This must be
done simultaneously for all sites (≈50,000) in the East African region in a multiple testing framework designed to
ensure a control over a meaningful combined measure of statistical Types I and III errors.
In this paper, we will first be framing the research question as a heuristic multi-objective temporal assignment
problem, in which better sub-regions were created than the arbitrarily chosen square grids in Clements et.al. (2014).
By using temporal assignments to create subregions, we will demonstrate that the testing procedure becomes more
powerful.  Also in this article, we provide theoretical proof that the mdFDR is still controlled under sub-region
independence.
**2.1 Temporal Assignment Problem Overview**



There is a wealth of research on assignment problems and specialized assignment problems that display
complicating constraints. Though the generalized assignment problem is solvable, once the number of dimensions
reaches 3, as in the formulation presented in this paper, this is no longer the case.
The multidimensional assignment problem was introduced by Pierskalla (1968) and a bibliography of multidi-
mensional assignment problems was prepared by Gilbert & Hofstra (1988). Miori (2011, 2008, 2014) used
assignment problems to model truckload routing problems and the Pollyanna gift exchange problem. Scheduling
medical residents with the temporal component was addressed by Franz & Miller (1993). Bandelt, Et al. (1994,
2004) addressed multi-dimensional assignment problems with decomposable costs. The three-dimensional
assignment problem was applied to teaching schedules by Frieze & Yadegar (1981) and Balas & Saltman (1991).
Multidimensional approximation was applied to capacity expansion problems by Troung & Roundy (2011).
Lagrangian Relaxation was applied to a multi-dimensional assignment problem arising from multi-target tracking by
Poore & Rijavec (1993). Multi-tracking data was also addressed by Robertson (2001).
Approximations to the multi-dimensional assignment problem were generated by Kuroki & Matsui (2007), Gutin,
Et al. (2008), Krokhmal, Et al. (2007), and Karapetyan &Gutin (2011). The multi-objective assignment problem
seeking solutions to the assignment problem in the face of additional objectives using efficient sets was posed by
White (1984). A weighting function approach has also been applied to multi-objective (multicriteria) problems with
conflicting objectives by Phillips (1987).
Agricultural planning problems have been addressed by Samuelson (1952), Takayama (1964), Norton & Scandizzo
(1981), Kutcher & Norton (1982), Önal & McCarl, and Weintraub & Romero (2017). Multicriteria approaches to
agriculture decisions have also been applied by Gasson (1973), Harper & Eastman (1980), Wheeler & Russel
(1977), Hayashi (2000), and Romero & Rehman (2003).

**2.2 Land Use Optimization Overview**
The most basic methods in land use optimization involve limited enumeration of alternatives and developing metrics
to directly assess these alternatives. Landscape metrics addressing various land use goals were used by Kuchma, Et
al. (2013) to evaluate enumerated options for land use. A similar approach was proposed by Wang & Guldmann
(2015) to mitigate seismic damages in Taichung, Taiwan.
Heuristic methods and in sustainable land use were applied by Steward, Et al. (2004), Cao, Et al. (2011), Liu, Et al.
(2016) and Sahebgharani (2016). Genetic algorithms were presented Cao, Et al. (2012) and the Analytical
Hierarachy Process was utilized by Memarian, Et al. (2014). Multi-objective linear programming with sensitivity
analysis was found effective by Sadeghi, Et al. (2009) while Soil and Water Assessment Toll (SWAT) was
employed by Sunandar, Et al. (2014).





## 3 Data Description

East Africa spans a wide variety of climate types and precipitation regimes which are reflected in its vegetation
cover. To capture this, satellite imagery was collected over a sub-Saharan region of East Africa that includes five
countries in their entirety (Kenya, Uganda, Tanzania, Burundi and Rwanda) and portions of seven countries
(Somalia, Ethiopia, South Sudan, Democratic Republic of Congo, Malawi, Mozambique and Zimbabwe). This
roughly 'rectangular' region extends from 27.8°E to 42.0°E longitude and 15.0°S to 6.2°N latitude. Also included in
the region are several East African Great Lakes such as Lake Victoria, Lake Malawi and Lake Tanganyika.
Vegetative analysis in this region is of interest for a variety of reasons, including the importance of the region for
global biodiversity and the vulnerability of the region to climate change, deforestation of the Congo, urban
development, civil conflict, and agricultural practices.

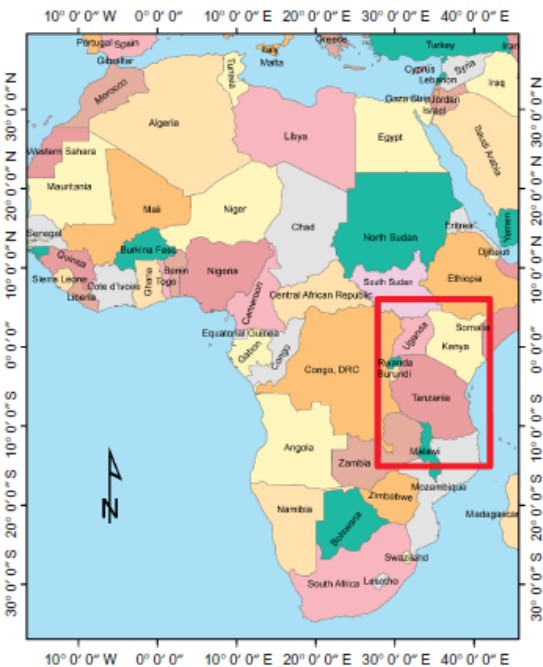


Figure 1 The study area, as indicated by the box.

The remotely sensed images were recorded twice a month from 1982–2006 and then converted to NDVI values.
Hence, the spatio-temporal data set consists of approximately 50,000 sites (pixels), each with 600 time series
observations (24 observations per year over 25 years). The satellite's resolution corresponds to each pixel spanning
an 8,000m × 8,000m grid of land, which we will refer to as a 'location.'  This Global Inventory Modeling and
Mapping Studies (GIMMS) data set is derived from imagery obtained from the Advanced Very High Resolution



Radiometer (AVHRR) instrument onboard the National Oceanic and Atmospheric Administration (NOAA) satellite
series 7, 9, 11, 14, 16 and 17. The NDVI values have been corrected by Tucker, Et al. (2005) for calibration, view
geometry, volcanic aerosols, cloud coverage and other effects not related to vegetation change.
All the negative NDVI values were consolidated to zero, as commonly done in vegetation monitoring, and re-scaled
the remaining values by 1,000. Negative NDVI values indicate non-vegetation areas, and so they are of no use in our
statistical analysis. Prior to the analysis, we examined the data for quality assurance and eliminated a small number
of pixels that were found to have several consecutive years with identical data values, which may be due to data
entry errors or machine malfunction.
When this data was first examined in Vrieling, de Beurs and Brown (2008), the percentage of pixels with the trend
test p-value less than $\alpha = 0.10$ was reported separately for positive and negative slopes. The reported results indicate
that much of the region has 'significant' vegetation change. For example, the cumulative NDVI indicator detected
44.2% of sites with p-values less than 0.10. However, this result fails to address the important statistical issue of
multiplicity when making these claims about significant vegetation changes and their directions simultaneously for
all the regions based on hypothesis testing.  Later, Clements, et. al. (2014) addressed the multiplicity issue by
proposing a 3-stage multiple testing procedure to control the mixed-directional False Discovery Rate (mdFDR), but
did so on subregions of East Africa that were not optimally formed.
The associated csv file for this analysis is the information generated from Clements, et al, (2014) which was the
initial starting point for this analysis.  It contains the following fields:
• site: Consecutive ID number, acting as a unique identified
• xcoord: pixel longitude
• ycoord: pixel latitude
• ndvi.avg: Overall pixel average NDVI from 1982 to 2006 (observations taken twice monthly)
• pval: Resulting p value from the Brillinger Trend Test (Brillinger, 1989)
• slp: Resulting slope from the Brillinger Tren Test (Brillinger, 1989)
• block: Block number – initial assignment was arbitrary
Using the algorithm below, followed by the multiple testing procedure, users may generate the revised and improved
block assignments.

### 4 Assignment Problem Formulation

We propose an assignment formulation to this problem, using these analysis results, with the goal of an improved
solution.  The object of the geographic assignment problem is to map each pixel within the satellite images to an
appropriate block based upon a target value for each block.  The block target values represent equal size ranges
within the overall range of the objective function values.  The objective function for the pixel assignment is the sum





of absolute difference between the pixel NDVI and the block target NDVI. The number of blocks is set objectively
and may be reset for each assignment problem solution generated.
Note that pixels may be formed entirely of water; these pixels have been assigned arbitrarily high NDVI values to
effectively eliminate them from consideration in the block assignments. A 'water block' with an arbitrarily high
target value ensures that all of these pixels may be assigned to blocks.
The objective of the pixel assignment problem is to minimize the NDVI difference function. Let m = the number of
pixels, let n = the number of blocks, and let T = the number of time periods. The decision variable $x_{ij}^k$ is a binary
variable that represents the assignment, or lack of assignment, of pixel i to block j at time k. The constraints
formulated ensure that each pixel is assigned to a block, during each period of time. The formulation in Eq. (1) – (3)
follows the notation.

| $x_{i,j}^k$ | Decision variable $\in (0,1)$ $i = 1, \cdots, m; j = 1, \cdots, n; \ k = 1, \cdots, T$ |
|---|---|
| $N_{i\cdot}^k$ | Pixel $i$ NDVI score for time $k$: $i = 1, \cdots, m; \ k = 1, \cdots, T$ |
| $N_{\cdot j}^k$ | Block $j$ NDVI target for time $k$: $j = 1, \cdots, n; \ k = 1, \cdots, T$ |

**Table 2 Assignment Problem Notation.**

$$Minimize \sum_k \sum_j \sum_i \left| N_{\cdot j}^k - N_{i\cdot}^k \right| \cdot x_{i,j}^k \qquad (1)$$

Subject to:
$$\sum_i \sum_j x_{i,j}^k = 1 \ for \ k = 1, \cdots, T \qquad (2)$$

$$x_{i,j}^k \in (0,1) \qquad (3)$$

The binary decision variables utilize three indices, rendering the problem NP hard. We therefore propose and
employ a heuristic approach that relies heavily on dynamic programming.
**5 Assignment Problem Solutions**
**5.1 Lagrangian Relation**
Restatement of the pixel assignment problem as a Markov Process will facilitate alternative solution methodologies.
We present a Lagrangian relaxation of the formulation and introduce a Lagrangian multiplier ($\varphi_k$) for the single
constraint to be relaxed in each time period k = 1, $\cdots$, T. We include a simplifying assumption that the penalty is
constant over all time periods and is denoted as $\varphi$  The revised formulation is presented in Eq. (4) - (5).

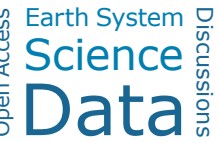

$$Minimize \sum_k \sum_j \sum_i |N_{\cdot j}^k - N_{i\cdot}^k| \cdot x_{i,j}^k + \sum_k \varphi \left( \sum_j \sum_i x_{i,j}^k - 1 \right) \qquad (4)$$

Subject to:
$$x_{i,j}^k \in (0,1) \qquad (5)$$

A dynamic programming formulation may now be presented using the relaxed formulation.
**5.2 Dynamic Programming Formulation**
The pixel assignment decisions may be made in stages, and while the outcome of each decision is not fully
predictable, it can be observed before the next decision is made. We begin the dynamic programming formulation
by organizing the problem into a tree structure (Fig. 2) reflecting pixels and levels (time increments). Each level of
the tree corresponds to a time increment, beginning with time 0 which represents the first satellite images retrieved
within the data set and ending at the final images at time T-1 and the pixels in each level number from 1 to m. The
tree provides a discrete-time dynamic system.

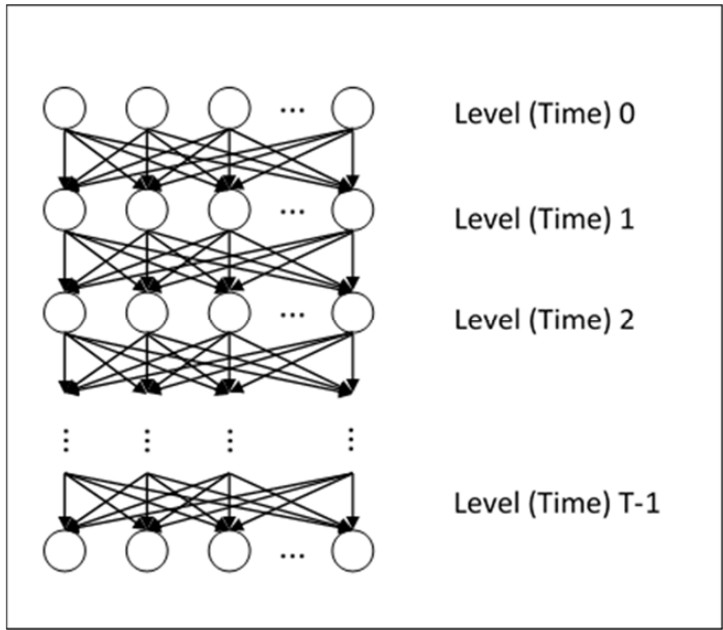


294                 Figure 2 General Tree Structure.






An additive value function reflects both present cost of each pixel assignment to a block, and potential future cost of
all pixel assignments to blocks (expected cost-to-go). Block NDVI targets must be established in order to match
pixels to blocks. Initialization of these targets is accomplished by evenly distributing the range of NDVI values
across n candidate blocks. Recall that the NDVI values ranges between 0 and 1000, resulting in block targets
starting at zero with an increment $1000/n$ up to 1000.
To calculate expected cost-to-go, we must also identify and calculate transition probabilities. In doing so, we
consider only the current level (time period). The Markov Property (6) allows us to omit consideration of the
probabilities of the path leading to the current level. The tree may now be viewed as a finite Nonhomogeneous
Markov Process with transition probability matrix $P^{(k)}$ representing transitions at any level.
$$P(X_{k+1} = x_{k+1}|X_0 = x_0, \dots, X_k = X_k) = P(X_{k+1} = x_{k+1}|X_k = x_k) \qquad (6)$$

The objective of the dynamic programming formulation is the minimization of the sum of cost at the current stage,
and the cost-to-go (the best case to be expected from future stages). The notation required for the formulation
follows.

| | |
|---|---|
| $A(k+1, k)$ | Available pixels at level (time) $k+1$, depends on pixel chosen at level $k$ |
| $m_{k+1}$ | Cardinality of $A(k+1, k)$ (the number of pixels available at level $k+1$, depends on pixel selected at level $k$) |
| $s(k)$ | The pixel chosen at level $k$ |
| $\boldsymbol{P}^{(k)}$ | Transition probability matrix at level $k$ |
| $\boldsymbol{P}^{(k)}_{i,j}$ | Transition probability of moving from pixel $i$ to pixel $j$ at level $k$ |
| $C_{s(k),j}$ | Cost of adding node $j$ after the pixel chosen at level $k$ |
| $U(i, k)$ | The number of unassigned pixels if we choose pixel $i$ at level $k$ |
| $f(i, k)$ | Expected cost-to-go if we choose pixel $i$ at level $k$ |
| $f(1,0)$ | Initialize to 0 |
| $\varphi$ | Pixel assignment penalty |


Pixel assignments to blocks may begin at any pixel in level 0 of the tree and end at any pixel in level T-1. All pixels
must be assigned to a single block but individual blocks need not have pixels assigned to them. Let z be the
candidate block.
$$f(s(k), k) = \min_{z \in A(k+1,k), \varphi} \left\{ C^k_{s(k),z} + \boldsymbol{P}^{(k)}_{i,j} \varphi U(s(k), k) + f(z, k+1) \right\} \qquad (7)$$

$$C^k_{s(k),z} = \left| N^k_{\cdot z} - N^k_{s(k) \cdot} \right| \qquad (8)$$

Though this approach resolves issues with the original assignment formulation, it necessitates the calculation of
transition probability matrices $\left(P^{(k)}\right)$ at each level. Transition probabilities are dependent on the number of blocks
chosen, and the ability to statistically characterize the changes in vegetation in the pixels over time. With as few as





100 blocks, the probabilities would have a very small order of magnitude and an expectation of high levels of
inaccuracy, resulting in a lack of ability to detect meaningful differences.  We present a heuristic, rooted in dynamic
programming principles to render an efficient and useful solution to the pixel assignment problem.

**5.3 Recursive Heuristic Procedure**

Due to the original assignment problem being NP hard, and the dynamic programming approach resulting in
extreme computational and structural complexity, we introduce a heuristic method that leverages knowledge gained
in the assignment and dynamic programming approaches.  This heuristic also leverages the previous research
completed in controlling the mdFDR.
The heuristic procedure was initialized with the 150 blocks used in Clements et. al (2014) and 56,355 total pixels,
and utilized the previously calculated slopes and resulting p-values from monotonic trend tests.  Rather than
assigning the pixels to blocks over the duration of the 25-year span of the data collection as the assignment
formulation would, this approach focused on assignment at the final observations in the 25th year but the use of
slope and p-value allowed the approach to reflect the trends that occurred over time.  This same approach could be
used at any time during the study, reflecting all previous data.
The heuristic performance metric, like the objective function in the pixel assignment problem, required the
calculation of block values corresponding to the pixel values.  The metric leverages the initial random blocks by
including the block average NDVI, the block average slope, the block average p-value, and the slope change
indicator variable.  Notation is introduced in Table 3, followed by the formulation of the performance metric.

| | |
|---|---|
| $y_{fg}$ | Block assignment $\in (0,1)$ $f = 1, \cdots, m; g = 1, \cdots, n$ |
| $I_{fg}$ | Slope change Indicator variable |
| $N_{f\cdot}$ | Pixel $i$ NDVI score at final observation: $i = 1, \cdots, m$ |
| $NR_g$ | NDVI range for block $g$ |
| $S_{f\cdot}$ | Pixel $i$ slope over time: $i = 1, \cdots, m$ |
| $SR_g$ | Slope Range for block $g$ |
| $P_{f\cdot}$ | Pixel $i$ p-value over time: $i = 1, \cdots, m$ |
| $PR_g$ | p-value range for block $g$ |
| $w_d$ | Weight for scoring factor $d$: $d = 1, \cdots 4$ |

336                    **Table 3 Heuristic Metric Notation.**

Let f = pixel number and let g = block number and let
$$y_{fg} = \begin{cases} 1 & \text{if pixel } f \text{ is assigned to block } g \\ 0 & \text{if pixel } f \text{ is not assigned to block } g. \end{cases}$$


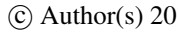


Development of the performance metric required definition of the block average values for NDVI, slope and p-value
shown in Eq (9) – (11). In addition, the indicator parameter, signaling slopes of opposite sign is shown in Eq. (12).

$$\bar{N}_{.g} = \sum_{f \ni pixels \in block\ g} \frac{N_{fg}}{n} \quad \forall g = 1, \cdots, n \tag{9}$$

$$\bar{S}_{.g} = \sum_{f \ni pixels \in block\ g} \frac{S_{fg}}{n} \quad \forall g = 1, \cdots, n \tag{10}$$

$$\bar{P}_{.g} = \sum_{f \ni pixels \in block\ g} \frac{P_{fg}}{n} \quad \forall g = 1, \cdots, n \tag{11}$$

$$I_{fg} = \begin{cases} 1 \text{ if } sign(S_{f.} * \bar{S}_{.g}) \text{ is negative } \forall f \ni pixels \in block\ g \\ 0 \text{ if } sign(S_{f.} * \bar{S}_{.g}) \text{ is positive } \forall f \ni pixels \in block\ g \end{cases} \tag{12}$$

The minimum value of the performance metric in Eq. (13) determines the highest quality heuristic solution. Pixels
whose current assignment leaves them on the border between blocks are evaluated. The metric is calculated for their
incumbent (current) assignment and their prospective assignment(s). The pixel is then assigned to the block yielding
the lowest value of the metric. As pixels are reassigned, newly exposed border pixels are evaluated in the same
fashion. This procedure continues until all border pixels belong to the block with the best fit.

$$w_1 \frac{|N_{f.} - \bar{N}_{.g}|}{NR_g} + w_2 \frac{|S_{f.} - \bar{S}_{.g}|}{SR_g} + w_3 \frac{|P_{f.} - \bar{P}_{.g}|}{PR_g} + w_4 I_{fg}$$

$$\forall \text{ pixel } f = 1, \cdots, m; \text{ bordering block } g = 1, \cdots, n \tag{13}$$

The dynamic programming concept of forward and backward passes has been adapted for the heuristic to
compensate for directional bias in the results. In this way, all border pixel assignments may be evaluated in all
directions. Four starting points and starting directions are identified in Fig. 3. Fig. 4 shows the four passes to be
completed for the first starting direction (upper left-hand corner). The first two passes are the forward direction
evaluation and the second two passes are the backward direction evaluation. These same four passes are adapted for
each starting point/direction, with the first pass always corresponding to the starting position.

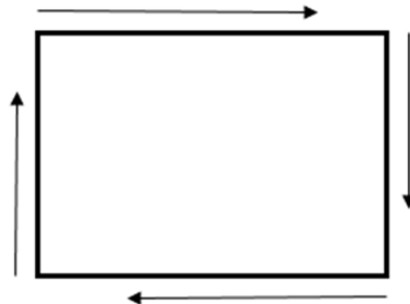




Figure 3 Starting Directions for Evaluation of Pixel Assignments.

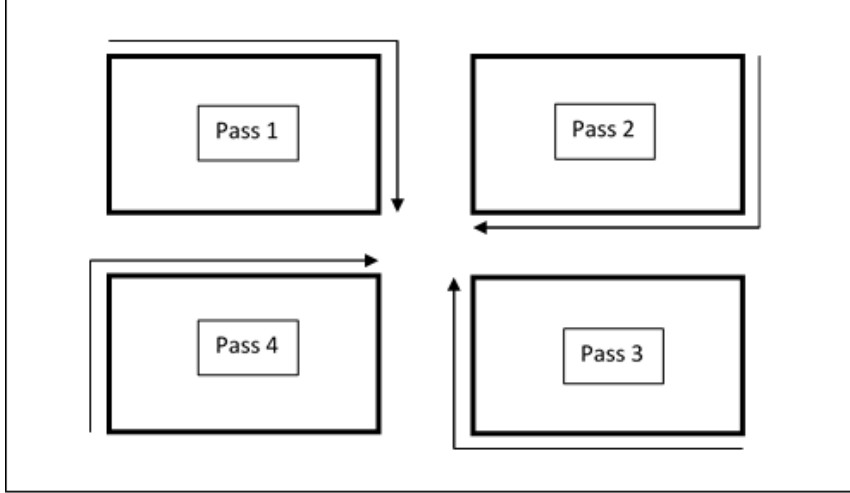


Figure 4 Forward-backward evaluation: Forward passes 1 and 2; Backward passes 3 and 4.

Implementation and validation of the heuristic was accomplished through the development of a program written in
the C programming language.
**6 Reassignment Model and Implementation**
An approach inspired by dynamic programming was utilized to find the best solution to the heuristic problem based
on weight factors that varied between 0 to 1, under the condition that $\sum_{i=1}^{4} w_i = 1$. Table 4 shows a subset of the
factor weight combinations that were examined. As seen in Table 4, selecting the solution with factor scores of
$w_1 = 1$, $w_2 = 0$, $w_3 = 0$, and $w_4 = 0$ generates the smallest value of the performance metric in Eq. (13). Since
factor 1 represents the NDVI average value at the final observation, this solution suggests performing pixel
reassignment based solely on NDVI information with no weight applied to factors such as slope and p-value. The
average score function of initial arbitrary square grid solution (calculated to be 0.1339) was compared to the
proposed reassignment solution (calculated to be 0.0998), and yielded an improvement of 25.5%.
[Table 4 near here]
The spatial map in Fig. 5 visualizes the initial arbitrary block assignment using square grids (left) compared to the
final solution (right) that gave the minimum value of the performance metric in Eq. (13). The contrast in maps
reveals how the solution to the pixel assignment problem created natural looking clusters of differing sizes. For
example, along some coastline areas, clusters are long and narrow. This is intuitive because NDVI values tend to be





similar along the coast where many areas are comprised of sand and rock. In other areas, clusters became circular
and cover vast areas of known deserts in the East African regions. Small clusters also exist in the solution and, after
investigating, we found that many of these clusters comprise of cities and urban areas that have little vegetation. It
is logical that such pixels should be reassigned into the same cluster.

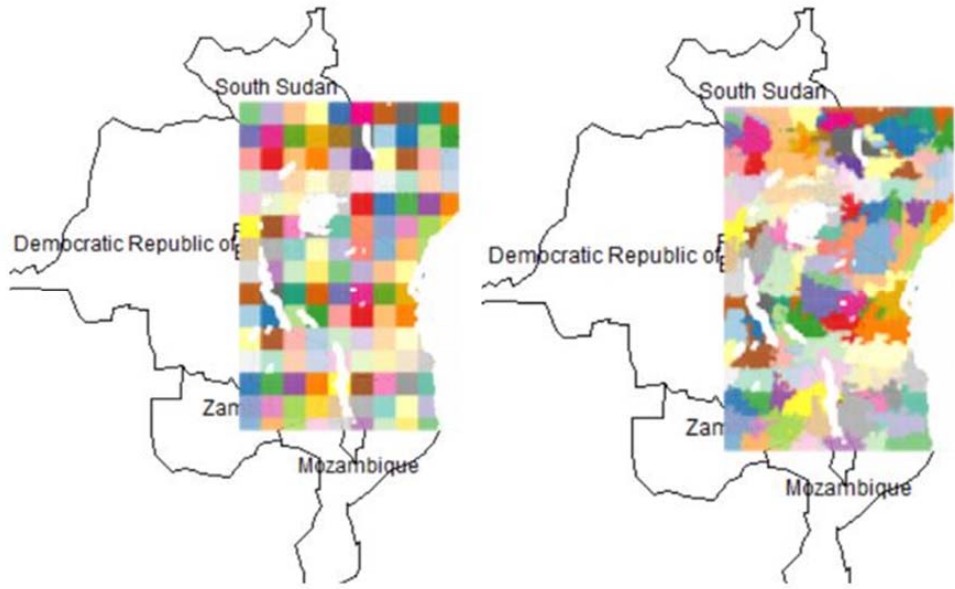


Figure 5 Initial arbitrary block assignment (left) compared to final solution (right).

An unbiased validation of the reassignment solution can be calculated using the average coefficient of variation for
the final pixel assignment and compare it to the initial square block assignment. The coefficient of variation (CV) is
a unit-less measure of spread that describes the amount of variability relative to the mean. CV is defined as the ratio
of standard deviation over the mean. Smaller values of CV indicate higher homogeneity of the clusters. The
average of cluster's coefficients of variations for our final pixel assignment solution is 11.762. This is a 27.4%
decrease compared to the average coefficient of variation for the original square blocks, which was 16.205. This is a
statistically significant difference in CV averages (p=0.000529), providing further evidence that the pixel
reassignment solution was able to increase the level of homogeneity within clusters. Having homogeneous clusters
is important when making large scale decisions about regions in East Africa that have experienced significant
vegetation trend changes.
**7 Multiple Testing Implementation and Results**





Now we can assume that the pixels in the East African region are divided into homogeneous subregions using
temporal assignments, as described above. Next, we summarize and apply the multiple testing procedure given in
Clements, et. al. (2014).
For notation, let m be the number of such subregions and $n_i$ be the number of pixels/locations in the $i^{th}$ subregion.
P-values at each location were calculated using a two-sided monotonic trend test at each location using the Brillinger
(1989) test. Specifically, we denote $\beta_{ij}$ as the monotonic trend parameter as defined in the Brillinger test for the $i^{th}$
subregion and $j^{th}$ location, where $i = 1, 2, ..., m, j = 1, 2, ..., n_i$. We also let $T_{ij}$ and $P_{ij}$ be, respectively, the test
statistic and the corresponding p-value for testing the null hypothesis $H_{ij}: \beta_{ij} = 0$ against its two-sided alternative
$H_{i1}: \beta_{ij} \neq 0$.
We apply Clements, et. al. (2014) suggestion of using a Bonferroni correction at each subregion, which combines
the p-values by calculating $P_i = n_i \min_{1 \leq j \leq n_i}(P_{ij})$. With $H_{ij}$ representing the null hypothesis corresponding to $P_{ij}$,
consider $H_i = \bigcap_{j=1}^{n_i} H_{ij}$ as the null hypothesis corresponding to $i^{th}$ subregion. We will test the $H_{ij}$'s against their
respective two-sided alternatives and detect the direction of the alternatives for the rejected hypotheses.
Specifically, we apply the procedure using α=0.05 in the following three steps:
*Multiple Testing Procedure Applied to Homogeneous Sub-regions:*
1)        Apply the BH method to test $H_i$, $i = 1, 2, ..., m$, based on their respective p-values $P_1, P_2, ..., P_m$ as follows:
consider the increasingly ordered versions of the $P_i$'s, $P_{(1)} \leq P_{(2)} \leq ... \leq P_{(m)}$. Find $S = \max\{i: P_{(i)} \leq i\alpha/m\}$.
Reject the $H_i$'s for which the p-values are less than or equal to $P_{(S)}$, provided this maximum exists, otherwise, accept
all $H_i$.
2)        For every $i$ such that $H_i$ is rejected at step 1, consider testing $H_{ij}$, $j = 1, 2, ..., n_i$ based on their respective p-
values $P_{ij}$, $j = 1, 2, ..., n_i$, as follows: reject $H_{ij}$ if $P_{ij} \leq S\alpha/mn_i$.
3)        For each rejected $H_{ij}$ in step 2, decide the direction of the monotonic trend to be the same as that of
$sign(T_{ij})$.
Step 1 and 2 identify first, the subregions and second, the locations with significant vegetation changes. The third
step allows one to make a more detailed analysis by identifying the directions in which these significant changes
have occurred. Impressively, this procedure controls the mdFDR at level α if the subregions are independent. A
mathematical proof of this is given in the Appendix.
The results of implementing this procedure to our homogenous subregions are shown in Fig. 6. Sites with a
significant increasing change in vegetation are plotted in green. Sites with significant negative vegetation change are
plotted in red. The nonsignificant sites are represented by white. Using the temporal reassignment to form



homogeneous subregions before implementing the multiple testing procedure detected 518 locations with significant
vegetation changes. Compared to the procedure in Clements, et. al. (2014) based on arbitrary square subregions,
this is an increase in 10 detected locations, which is indicative of a higher-powered testing procedure, while still
maintaining control over Type I and Type III errors.
Geographically, the results show increasing vegetation trends in the Northern hemisphere as well as coastal Eastern
Tanzania. Decreasing vegetation trends are mostly concentrated directly South of Lake Victoria. These findings are
consistent with historical evidence and other climate change investigations done in this region.

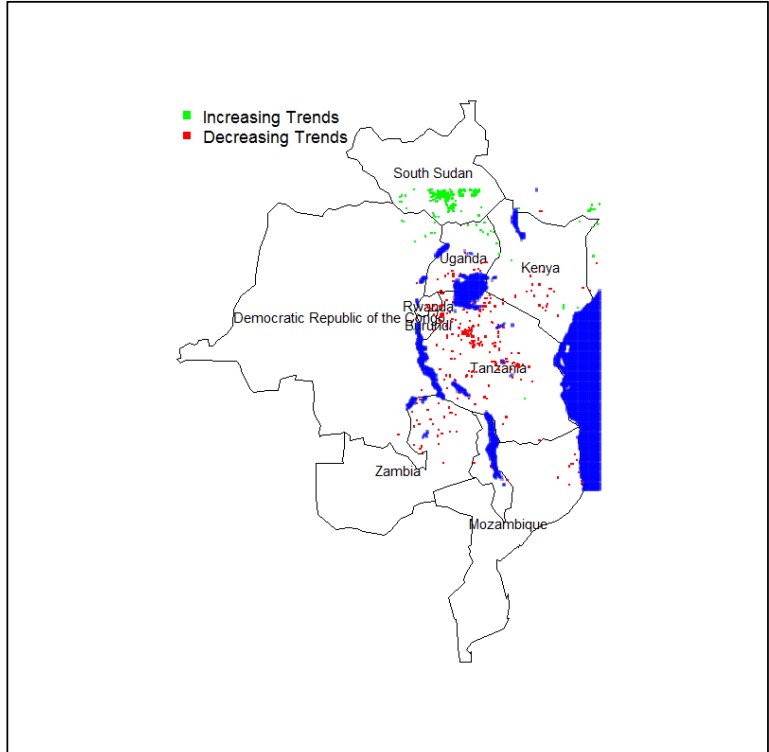


437          Figure 6 Pixels detected using the proposed heuristic reassignment solution with multiple testing procedures.

**8 Data Availability**
The data, titled "NDVI and Statistical Data for Generating Homogeneous Land Use Recommendations", may be
accessed through figshare. The link to the archives is: https://figshare.com/s/ed0ba3a1b24c3cb31ebf and the DOI is
https://figshare.com/articles/NDVI_and_Statistical_Data_for_Generating_Homogeneous_Land_Use_Recommendati
ons/5897581.
**9 Conclusions and Future Research**



It is important to consider neighboring pixel's vegetation when making costly land management decisions that
would potentially relocate East African populations of people, livestock and crops.  The motivation of this paper
stems from the opportunity to optimize the pixel assignments based on neighboring pixel data, rather than using
blocks in an arbitrary grid fashion, prior to using statistical methodologies to detect vegetation changes over regions
in East Africa.  Knowing information about the homogeneous cluster to which a particular pixel belongs can provide
valuable insights and improved methodologies.
Although we demonstrated our methodology using NDVI data, the procedure can be used for any spatial-temporal
data, even on finer scales. Overall, by using dynamic programming to formulate a multidimensional temporal
assignment problem implemented by the heuristic procedure, we were able to reassign pixels to adjacent clusters
based on similar NDVI values over time.  The results of this analysis create more homogeneous regions of East
Africa for decision makers to draw inferences regarding vegetation changes.  We have demonstrated a powerful tool
for homogeneous cluster creation of pixels undergoing land-cover change using temporal satellite data.
Efficient land use for economic sustainability and effective land use for environmental sustainability have become
very important topics addressed by Cole, Et al. (2000) and Duveiller, Et al. (2007).  This research may be directly
extended to consider additional characteristics of land and identify appropriate land use as in Usongo & Nagahuedi
(2008).  This is especially important when considering the inclusion of multiple land purposes: residential, farm,
riparian borders, industrial, commercial, etc.
Another avenue to explore in future research is to extend the proposed methodologies to other applications of spatio-
temporal data.  For example, monitoring and detecting transient sources in the night sky, specifically Type Ia
supernovae transients, is an area of astronomical research that receives much attention.  Spatio-temporal astronomy
data has spatial dependencies that exist between pixels in astronomical images, which is well suited for a
multidimensional temporal reassignment to create homogenous clusters.
With extension of this work to other special problems, finding optimal weights will become important and relevant
work.  Though the pixel assignment problems ultimately unveiled the appropriate weights through an iterative
approach, problems with extended criteria provide a greater challenge in determining appropriate or optimal
weights. We anticipate determination of optimal weights to be evaluated as future research as well.





**Appendix A**
**Proof**. We prove that the mdFDR is controlled at desired level $\alpha$, by borrowing some ideas in Clements, et. al.
(2014). Let R be the total number of $H_{ij}$'s that have been rejected, and V and U, respectively, be the numbers of
Types I and III errors that occurred out of these R rejections. Then

$$\text{mdFDR} = \text{FDR} + \text{dFDR} = E\left(\frac{V+U}{\max\{R,1\}}\right), \tag{A1}$$

since FDR $= E\left(\frac{V}{\max\{R,1\}}\right)$ and dFDR $= E\left(\frac{U}{\max\{R,1\}}\right)$ is the directional FDR. Let us consider using $H_{ij}$ as an indicator
variable with $H_{ij} = 0$ (or 1), indicating that Brillinger's null hypothesis $H_{ij}: \beta_{ij} = 0$ is true (or false). Then,

$$V = \sum_{i=1}^{m} \sum_{j=1}^{n_i} I\left(H_{ij} = 0, P_{ij} \leq S\alpha/mn_i\right) \tag{A2}$$

where $S$ is the number of significant subregions in the first stage of the procedure. Hence,

$$\text{FDR} = E\left(\frac{V}{\max\{R, 1\}}\right) \tag{A3}$$

$$= \sum_{i=1}^{m} \sum_{j=1}^{n_i} E\left(\frac{I\left(H_{ij} = 0, P_{ij} \leq S\alpha/mn_i\right)}{\max\{R, 1\}}\right) \tag{A4}$$

$$\leq \sum_{i=1}^{m} \sum_{j=1}^{n_i} I(H_{ij} = 0)E\left(\frac{I\left(P_{ij} \leq \frac{S\alpha}{mn_i}\right)}{\max\{S, 1\}}\right) \tag{A5}$$

since $R \geq S$ [borrowing the idea from Guo and Sarkar (2012)]. Let $S^{(-i)}$ be the number of significant subregions that
would have been obtained if we had completely ignored the $i^{th}$ subregion and applied the first-stage BH method to
the rest of the $m - 1$ subregion $p$-values using the critical values $\frac{i\alpha}{m}, i = 2,3, \dots, m$. Then, it can be shown that

$$\frac{I(P_{ij} \leq S\alpha/mn_i)}{\max\{S, 1\}} = \sum_{s=1}^{m} \frac{I\left(P_{ij} \leq \frac{s\alpha}{mn_i}, S = s\right)}{s} \tag{A6}$$

$$= \sum_{s=1}^{m} \frac{I\left(P_{ij} \leq \frac{s\alpha}{mn_i}, S^{(-i)} = s - 1\right)}{s} \tag{A7}$$

Since we assume the $m$ subregions are independent, taking expectation and inserting into FDR definition gives us





$$\text{FDR} \leq \sum_{i=1}^{m} \sum_{j=1}^{n_i} I(H_{ij} = 0) \sum_{s=1}^{m} \frac{1}{s} \frac{s\alpha}{mn_i} \Pr(S^{(-i)} = s - 1) \qquad (A8)$$

$$= \alpha \sum_{i=1}^{m} \frac{1}{mn_i} \sum_{j=1}^{n_i} I(H_{ij} = 0) \qquad (A9)$$

$$= \frac{\alpha}{m} \sum_{i=1}^{m} \pi_{i0} \qquad (A10)$$

where $\pi_{i0}$ is the proportion of true null hypotheses among the total $n_i$ null hypotheses in the $i^{th}$ subregion.

We now work with the dFDR. Let $\delta_{ij} = \text{sign}(\beta_{ij})$ representing the true sign of the Brillinger's monotonic trend
parameter $\beta_{ij}$ $j^{th}$ location in the $i^{th}$ subregion and $T_{ij}$ is the test statistic. Now, $U$ can be expressed as follows:
$$U = \sum_{i=1}^{m} \sum_{j=1}^{n_i} I\left(H_{ij} = 1, P_{ij} \leq \frac{S\alpha}{mn_i}, T_{ij}\delta_{ij} < 0\right) \qquad (A11)$$

from which we first have

$$\text{dFDR} = E\left(\frac{U}{\max\{R, 1\}}\right) \qquad (A12)$$

$$U = \sum_{i=1}^{m} \sum_{j=1}^{n_i} I(H_{ij} = 1) E\left(\frac{I\left(P_{ij} \leq \frac{S\alpha}{mn_i}, T_{ij}\delta_{ij} < 0\right)}{\max\{R, 1\}}\right) \qquad (A13)$$

Making arguments similar to those used for the FDR, we then have
$$\text{dFDR} \leq \sum_{i=1}^{m} \sum_{j=1}^{n_i} I(H_{ij} = 1) \sum_{s=1}^{m} \frac{1}{s} \Pr\left(P_{ij} \leq \frac{s\alpha}{mn_i}, T_{ij}\delta_{ij} < 0\right) \Pr(S^{(-i)} = s - 1) \qquad (A14)$$

Notice that $P_{ij} = 2[1 - \Phi(|T_{ij}|)]$, where $\Phi$ is the cumulative distribution function of the standard normal.
Therefore, assuming without any loss of generality that $\beta_{ij} > 0$ when $H_{ij} = 1$, we have, for such $H_{ij}$,
$$\Pr\left(P_{ij} \leq \frac{s\alpha}{mn_i}, T_{ij}\delta_{ij} < 0\right) \qquad (A15)$$



$$= \Pr_{\beta_{ij}>0}\left(|T_{ij}| \geq F^{-1}\left(1 - \frac{s\alpha}{2mn_i}\right), T_{ij} < 0\right) \qquad (A16)$$

$$= \Pr_{\beta_{ij}>0}\left(T_{ij} \leq -F^{-1}\left(1 - \frac{s\alpha}{2mn_i}\right)\right) \qquad (A17)$$

$$\leq \Pr_{\beta_{ij}=0}\left(T_{ij} \leq -F^{-1}\left(1 - \frac{s\alpha}{2mn_i}\right)\right) \qquad (A18)$$

$$= \frac{s\alpha}{2mn_i}.$$

The last inequality follows from the fact that, when $H_{ij} = 1$, the distribution of $T_{ij}$ is stochastically increasing in $\beta_{ij}$.
Continuing, we have
$$\text{dFDR} \leq \frac{\alpha}{2m}\sum_{i=1}^{m}\frac{1}{n_i}\sum_{j=1}^{n_i} I(H_{ij} = 1) = \frac{\alpha}{2m}\sum_{i=1}^{m}\pi_{i1} \qquad (A19)$$


where $\pi_{i1}$ is the proportion of false null hypotheses among the total $n_i$ null hypotheses in the $i^{\text{th}}$ subregion. Thus, we
combine and finally prove the desired result.
$$\text{mdFDR} \leq \frac{\alpha}{m}\sum_{i=1}^{m}\left(\pi_{i0} + \frac{1}{2}\pi_{i1}\right) = \frac{\alpha}{m}\sum_{i=1}^{m}\left(\frac{1 + \pi_{i0}}{2}\right) \qquad (A20)$$




Author Contribution
Nicolle Clements completed all multiple testing evaluation and statistical analysis. Virginia Miori completed all
decision model development.  Virginia Miori developed the heuristic approach; the heuristic was refined by
Virginia Miori and Nicolle Clements.  The development of computer code to implement the heuristic was completed
by Brian Segulin.
**Competing Interests**
The authors declare that they have no conflict of interest.
**Acknowledgements**
The NDVI data set was collected as part of a Michegan State University research project, namely, the "Dynamic
Interactions among People, Livestock, and Savanna Ecosystems under Climate Change" project (funded by the
National Science Foundation Biocomplexity of Coupled Human and Natural Systems Program, Award No.
BCS/CNH 0709671).



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
