# Peer review of "Heuristic Approach to Multidimensional Temporal Assignment of Spatial Grid Points for Effective Vegetation Monitoring and Land Use in East Africa"

_Earth System Science Data, 2018_

## Referee Comment (RC1) · Anonymous Referee #1 · 10 Aug 2018

Review ESSD-2018-18, East Africa Land and Vegetation

Despite its provocative title, this paper does not qualify for nor belong in ESSD. Following from its self-professed keywords " Land Use, Mathematical Programming, Dynamic Programming, Multiple Testing, Spatial Data and Analysis, False Discovery Rate", the manuscript addresses mathematicians and statisticians but not earth system scientists. Essentially the authors want to demonstrate a better method to assign 2-D spatial patterns to terrestrial environments to improve detection and monitoring of land use change. Their basic premise - that improved mapping should enable improved detection - seems entirely plausible. One hopes they would have combined knowledge of land vegetation patterns and remote sensing with statistical techniques for delineating land use patterns. Instead they fail completely on vegetation, demonstrate an appalling ignorance of remote sensing, and may or may not have made appropriate application of mathematical tools. They proclaim - in line 172 on page 6 - that they "provide theoretical proof that the mdFDR is still controlled under sub-region independence"; they should submit 'proofs' elsewhere. If they want to contribute substantially, via publication in ESSD, to understanding of land use change and its social implications, they need to offer a much different, much improved approach. They wish to convince readers (page 18, lines 434, 435) that their "findings are consistent with historical evidence and other climate change investigations done in this region" but they fail to provide such evidence from their own work and can not, evidently, cite any third-party confirming evidence.

In generating this review I assembled a long list of mistakes and mis-statements, in many sections several per paragraph. The authors have not met journal expectations of clear precise language or of well-documented assertions. Because I recommend that they discard the present manuscript - through formal rejection by the journal or withdrawal by the authors - I will not inflict those many pages of comments and objections. If the authors want to start fresh, I make the following suggestions.

1) Start by analysing a region of known land use distribution and patterns. Make a choice that provides desired combinations of topographic variation, record duration and confirmed accuracy; many examples exist. A quick search in ESSD reveals: the UK (Wood et al., 40 years of country-side surveys, https://doi.org/10.5194/essd-10-745-2018); the continental USA (e.g. Chen et al., 80 years of plantation forest data, https://doi.org/10.5194/essd-9-545-2017);) Siberia (Ottle et al. https://doi.org/10.5194/essd-5-331-2013); agricultural regions in Korea (Seo et al. https://doi.org/10.5194/essd-6-339-2014); even a global satellite-based view covering nearly 25 years from which these authors might select interesting regions (Li et al. https://doi.org/10.5194/essd-10-219-2018). I suspect they could find good examples for mid-Atlantic Appalachia-to-coast regions.

2) Use - as most of the above studies will have done - standard land use and plant functional type terminology. Start with the IGBP land use categories, a quick google search will turn up hundreds of examples. For example: http://glcf.umd.edu/data/lc/. By adopting standard terminologies these authors will access a wide variety of pre-existing work while making their own work accessible and comparable to many subsequent users. Standard easily-accessible PFT (plant functional type) terminologies also exist.

3) Demonstrate your block assignment techniques on one of these known land use maps. Document improvements (?) relative to standard gridded approaches based on, again, existing well-described land use categories (from #2 above).

4) Compare outcomes based on known land use categories with remotely sensed parameters, specifically NDVI. Essentially, use the IGBP land use categories in test mode as a transfer standard to NDVI.

5) Get a thorough education in remote sensing of land surfaces. The current manuscript reveals a serious deficiency in this regard: uninformed and out-of-date. For this particular data set the authors want Pinzon and Tucker 2014 (Remote Sensing 2014, 6(8), 6929-6960; doi:10.3390/rs6086929) rather than Tucker et al. 2005); they should give particular attention to Panel E of Figure 4. Demonstrate, by narrative and perhaps as well by including remote sensing colleagues, that you appreciate the long, detailed and well-documented caution about quantitative application of NDVI. Demonstrate a clear understanding of the many factors that confuse or even obscure NDVI. Understand why readers and users will wonder greatly at a supposed average value of NDVI expressed as 575.8716664 (row 10 column D of the .csv file): ten (!!) significant figures? Many knowledgeable readers would rather expect to see $0.6 \pm 0.1$!

This reviewer had to generate my own frequency distribution of average NDVI (column D, rescaled by dividing by 1000 and rounding back to 2 significant figures) to convince me that the data represented NDVI; the resulting distribution differs substantially from Figure 4 Panel E of Pinzon and Tucker already mentioned. A subsequent improved version of this manuscript will need to demonstrate serious understanding and restraint in manipulation of satellite remote sensing data. Abundant literature exists! The authors could start by looking closely at Du et al., A global satellite environmental data record derived from AMSR-E and AMSR2 microwave Earth observations in ESSD (https://doi.org/10.5194/essd-9-791-2017).

6) Finally, if warranted by all previous steps, demonstrate the utility of your improved, tested and validated mapping technique to East Africa. Include an uncertainty assessment, which, not least, addresses the use of NDVI in that case due to absence, weakness or rapid change of land use categories.

In a substantially revised and improved manuscript, a reader would expect clear and precise language, more documentation of local conditions and changes, and much less jargon about "gift exchange problems" (page 7 line 150).

If the authors want to understand current issues in land use change they should read the recent summaries of land use modelling issues available in the CMIP special issue of GMD (e.g. https://doi.org/10.5194/gmd-9-2973-2016 and https://doi.org/10.5194/gmd-9-2809-2016). The authors clearly need assistance with land use terminology! They use, variously, "vegetation life cycles", "vegetation trends", "vegetation structure" and "land cover trends". These terms and phrases convey independent and distinct concepts to terrestrial ecologists but the authors use them casually without recognising the confusion they sow.

Abstract, line 25: data link does not work. ESSD generally does not use figshare, finding it unreliable. Many alternatives exist: 4TU, Pangaea, Zenodo, etc. Suggest you deposit the data at a reliable doi-based repository with a good browse and search capability. A figshare url does not qualify as a permanent identifier.

The authors clearly imply (page 9, line 241) that the data set provided (as .csv) derives in fact from previous work, e.g. Clements et al. 2014. Should an ESSD reader conclude that these authors have submitted a manuscript to a data publishing journal based on someone else's data? Unacceptable (and scarcely believable) if true!

---

## Editor Comment (EC1) · D. J. Carlson (Editor) · 12 Sep 2018

These comments verbatim from a second reviewer:

"The paper entitled "Heuristic Approach to multidimensional temporal assignment of spatial grid point for effective vegetation monitoring and land use in East Africa" by V.M. Miori et al seeks to develop and test a methodology to describe trends of vegetation changes in East Africa based on NDVI data. The work is presented as an improvement over similar techniques in that it reduces false identifications of changes and at the

same time takes into account spatial information from the NDVI data.

The paper is well written, despite some obvious typos, and the methodology is clearly described. The overall motivation for a better approach in developing a successful detection mechanism for vegetation changes is well grounded. The paper layout is reasonable, although the Literature Review section (Section 2) is not clear. I am personally not familiar with multiple testing techniques, and the description in Section 2 is problematic as it does not explain how the previous techniques were used with geophysical data. It would be probably beneficial for the paper if the authors invited the collaboration of experts in agricultural, terrestrial biology, land and land use change science, in order to better address the physical understanding of the statistical methods employed.

Furthermore, it is unclear how, from a well established trend of past changes in vegetation, one can predict changes in the future, purely based on one index (NDVI). Trends in vegetation are a result of climate change, but also natural variability, local weather and microclimate, as well as anthropogenic intervention on the local level. There are vegetation changes associated with humidity, water supply, draughts, floods, precipitation, soil destruction, ground erosion, wildfires, winds, solar radiation availability, cloudiness, atmospheric pollution etc which in addition to land use change and agriculture may impact the vegetation changes. None of these factors is taken into account, nor is there a qualification how these factors are represented in the approach described here.

Lastly, I feel unequipped to judge the merit of the statistical technique proposed here, as it would have been better evaluated by a more specialized peer group in a similarly specialized journal. Co-author Clements has collaborated on a similar paper in the International Journal of Remote Sensing published in 2013. That journal would have been more relevant for this article perhaps. It is however an important point the authors must reconcile whether they decide to publish this paper elsewhere: how can trend detection that is based on past state of the system and which does not take into account climate change and extreme events or any other meteorological, climatic, biogeochemical factor that influences the local ecological balances, can provide reliable predictions of future trends?

I recommend that this paper is not published in ESSD at this stage unless both my major criticisms are addressed. "

---

## Editor Comment (EC2) · D. J. Carlson (Editor) · 14 Sep 2018

After conveying comments from the second reviewer, I read again both comments and the manuscript as submitted. Unfortunately I must agree with most comments from both reviews. As one reviewer points out and as the authors intend, improved 2-D discrimination of landscape features (including vegetation) should enhance our ability to monitor, quantify and perhaps predict landscape change, but the authors have failed to demonstrate that their mathematical approach produces realistic tools for landscape mapping. I resonate with cautions about quantitative applications of NDVI: one might

end with such correlations out of necessity for the region and the application but one should not start from such assumptions without substantial demonstration, exploration and validation for other regions where one in fact knows the landscape features and the record of change.

If this manuscript goes to a final decision, we will definitely reject for good abundant well-documented reasons; the reviews seem very clear in this regard.

For the authors I consul the alternate option of withdrawal. With sufficient motivation, time and resources they might follow the validation recipe suggested by one review. Or they might present the mapping algorithm itself to a technical remote sensing conference, to gain community feedback? Of course they have these and perhaps other options regardless, whether they withdraw the manuscript or the journal rejects it. I wonder if a withdraw action makes more sense in this case? I do not want to encourage a revision with reviews so seriously negative.